# Evolution of NIN and NIN-like Genes in Relation to Nodule Symbiosis

**DOI:** 10.3390/genes11070777

**Published:** 2020-07-11

**Authors:** Jieyu Liu, Ton Bisseling

**Affiliations:** 1Laboratory of Molecular Biology, Department of Plant Sciences, Graduate School Experimental Plant Sciences, Wageningen University & Research, 6708 PB Wageningen, The Netherlands; jieyu.liu@wur.nl; 2Beijing Advanced Innovation Center for Tree Breeding by Molecular Design, Beijing University of Agriculture, Beijing 102206, China

**Keywords:** NIN (NODULE INCEPTION), NLP (NIN-like Proteins), evolution, root nodule symbiosis, legume, actinorhizal-like plants

## Abstract

Legumes and actinorhizal plants are capable of forming root nodules symbiosis with rhizobia and *Frankia* bacteria. All these nodulating species belong to the nitrogen fixation clade. Most likely, nodulation evolved once in the last common ancestor of this clade. NIN (NODULE INCEPTION) is a transcription factor that is essential for nodulation in all studied species. Therefore, it seems probable that it was recruited at the start when nodulation evolved. NIN is the founding member of the NIN-like protein (NLP) family. It arose by duplication, and this occurred before nodulation evolved. Therefore, several plant species outside the nitrogen fixation clade have NLP(s), which is orthologous to NIN. In this review, we discuss how NIN has diverged from the ancestral NLP, what minimal changes would have been essential for it to become a key transcription controlling nodulation, and which adaptations might have evolved later.

## 1. Introduction

Nitrogen-fixing root nodule-formation is a special property of some plant species. All these species belong to the nitrogen fixation clade (NFC), which is composed of four orders: Fabales, Rosales, Cucurbitales and Fagales [1]. Nodulation occurs abundantly within the Fabales (legumes). Legumes establish nodule symbiosis with Gram-negative bacteria belonging to different genera, which collectively are named rhizobium [2]. In the other three orders, nodulation is more rare. In general, nodulation in these orders is induced by Gram-positive *Frankia* bacteria [3]. The plants that form nodules with *Frankia* are named actinorhizal plants. The exception is the genus Parasponia (order Rosales), which forms nodules with rhizobia [4].

Nodule types and modes of infection within the NFC are diverse. Nevertheless, recent studies indicate that a common ancestor of the NFC gained the nodulation ability [5,6,7]. Later during evolution, many species within the NFC lost the nodulation ability. Interestingly, phylogenomic analyses revealed that the transcription factor NIN (NODULE INCEPTION) that is essential for nodulation became a pseudogene, or was lost in most non-nodulating species of the NFC [5,6]. In this review, we will discuss the evolution of NIN and its related NIN-like proteins in relation to nodule symbiosis.

Root nodule-formation involves intracellular infection, nodule organogenesis, and a negative feedback mechanism that controls the number of nodules. Strikingly, the transcription factor NIN has been shown to play an indispensable role in all these processes [8,9,10,11,12,13]. For infection, the most common and advanced way is when bacteria attach to the root hair tip and stimulate the root hair to curl [14]. In this way the bacteria are entrapped in an enclosed cavity. By the deposition of new plant cell wall material, and invagination and growth of the plasma membrane, a tube like structure, the infection thread, is formed, which guides bacteria into the plant [14,15]. Alternatively, the bacteria can enter the roots without forming such infection threads, for example, through intercellular infection or crack entry [16,17].

In several legumes, the role of NIN in forming infection threads has been well studied. Loss of function of *NIN* leads to excessive root hair curling, and infection thread formation is blocked [8,9,10]. Similarly, NIN also most likely plays a role in infection of the actinorhizal plant, as it has been shown to be required for *Frankia*-induced root hair deformation in *Casuarina glauca* (Casuarina) [11]. This suggests that the role of NIN in infection thread formation is conserved in both legumes and actinorhizal plants.

In the model legume *Medicago truncatula* (Medicago), nodule formation starts with the mitotic activation of pericycle cells, and this is followed by divisions in cortical and endodermal cells [18]. The divisions in pericycle- and endodermis-derived cells stop at an early stage of nodule primordium formation, whereas divisions in the cortical cells persist. The cells derived from the cortex become infected and form the central tissue with infected cells. Cells derived from the cortex also form the peripheral tissue, including nodule vascular bundles [18].

The formation of actinorhizal nodules also starts with mitotic activation of the pericycle and cortical cells [19,20]. During nodule primordium formation, the pericycle-derived cells remain mitotically active. Previously, it has been described that these cells form the (complete) nodule [19,20]. However, it has been shown recently that these pericycle-derived cells only form the nodule vasculature [7]. Furthermore, during actinorhizal nodule primordium formation, the cortex-derived cells form the tissue with infected cells. Nodule formation in *Parasponia andersonii* (Parasponia) is similar to that in actinorhizal plants (actinorhizal-like) [7]. So, the major difference between legume and actinorhizal(-like) nodule organogenesis is the origin of the vascular bundle. In Medicago, it has been shown that a mutation in *NOOT1* causes a homeotic switch in the formation of the nodule vasculature, as it becomes actinorhizal-like since it is formed from pericycle cells that remain mitotically active [7,21].

Legume nodules can be divided in indeterminate and determinate nodules. Indeterminate nodules have a persistent meristem at their apex. This is similar to actinorhizal(-like) nodules. Due to their indeterminate growth, their tissues are of grade age, with the youngest cells near the meristem and the oldest in the part proximal to the root. NIN has been shown to be essential for both determinate [e.g., *Lotus japonicus* (Lotus)] [8] and indeterminate [e.g., Medicago and *Pisum sativum* (pea)] [9,10] nodules, as well as actinorhizal(-like) (e.g., Casuarina and Parasponia) [11,12] nodules. This indicates a common role of NIN in the formation of different types of nodules. 

Both infection and nodule organogenesis are initiated upon perception of signal molecules from the bacteria [22]. Most rhizobia secrete lipo-chito-oligosaccharides, called Nod factors (NFs), whereas the nature of the signals secreted by *Frankia* is not known. NFs activate a signaling pathway which is shared with the more ancient arbuscular mycorrhizal symbiosis, and NIN is the first induced transcription factor that distinguishes the rhizobium-activated responses from that of arbuscular mycorrhizae [23]. Although the nature of the *Frankia*-secreted signal is not clear, this common signaling pathway has also been shown to be required for actinorhizal(-like) nodule formation [24].

To balance costs and benefits during nodule symbiosis, plants developed a mechanism called autoregulation of nodulation (AON) by which nodule number is controlled [25,26,27,28,29,30,31]. It involves a communication between root and shoot. The signals that are sent to the shoot are CLAVATA3/ENDOSPERMSURROUNDING REGION (CLE) peptides [13,28,29,32]. For example, in Lotus, these are CLE-RS1 and CLE-RS2, and NIN directly induces their expression by binding to the NIN-binding sequence (NBS) in their promoters [13]. Upon perception of CLE peptides in the shoot, signals are sent to the root, resulting in reduced *NIN* expression, and so expression of targets will be reduced [13]. Thus, NIN plays a central role in the feedback loop, which ensures the formation of an optimal number of nodules.

NIN is a founding member of a small gene family called the NIN-like proteins (NLP) [33]. The studies on paralogues of NIN showed that they play an essential role in regulating nitrate-induced responses (reviewed in [34]). Interestingly, studies of Lotus showed that the expression of *CLE* genes is induced not only by rhizobia, but also by the application of nitrate [29,35]. One NLP (NRSYM1) can directly activate *CLE-RS2* expression in response to nitrate [35]. This suggests that the nitrate-induced block of nodulation shares common elements with AON, and NIN is partially functionally equivalent with NLPs. The latter is also supported by Lin et al. [36], which demonstrates that in Medicago NLP1 interacts with NIN to mediate nitrate inhibition of nodulation, and both NIN and NLP1 bind directly to the *CYTOKININ RESPONSE 1* (*CRE1*) promoter. NIN orthologues have also been shown to be present in species outside the NFC, like in *Solanum lycopersicum* (Tomato) and *Arabidopsis thaliana* (Arabidopsis) [11]. This indicates that NIN might be recruited in nodulation, based on its original function.

In this review, we will discuss possible evolutionary events underlying the recruitment of *NIN* in nodule symbiosis, based on comparing the NIN and NLPs of legumes, actinorhizal-like plants and non-nodulating species.

## 2. Phylogenetic Analysis of NIN

In Figure 1, a phylogenetic tree is shown, including NIN and NLPs from legumes, actinorhizal plant species, and monocot and dicot species outside the NFC. NIN and NLPs are divided into three orthogroups. Group 1 contains the NIN and NLPs of dicots, and is divided into two subgroups. One contains orthologues of NIN, and is named NIN subgroup. The other contains orthologues of MtNLP1 and is named NLP1 subgroup. Most likely these two subgroups are the result of a duplication that occurred before the NFC evolved, but after dicots and monocots separated. As a result, NIN orthologues occur in dicot species that are not within the NFC. Tomato and Arabidopsis NIN orthologues have been included in Figure 1. As these species have maintained a NIN orthologue, it strongly suggests that they have an essential non-symbiotic function. When NIN was recruited in the nodulation process, it most likely lost this non-symbiotic function. This is supported by the fact that most species within NFC that have lost nodulation also lost a functional *NIN* [5,6].

A comparison of *NIN* with their orthologues inside and outside the NFC, as well as with *NLPs* in the *NLP1* subgroup, will provide insight into the changes in the ancestral *NIN* that were essential for it to become a key regulator in nodulation. We will discuss these in the following paragraphs, and will focus on the properties of the proteins as well as the regulation of expression.

## 3. Evolutionary Adaptations in the NIN Promoter to Serve in Nodule Formation

*NIN* is a nodule specifically expressed in all studied species [6,8,9,10,11,12,37]. In contrast, most *NLPs* are constitutively expressed [36,38,39]. For example, in *Oryza sativa* (Rice), *Zea mays* (Maize), Arabidopsis and Medicago, *NLPs* are expressed in almost all organs, although with preferential expression in certain stages and tissues [36,38,39]. In legumes and actinorhizal(-like) plants, nodulation requires the common signaling pathway (see above), and in some species it has been shown that it induces the expression of *NIN* [6,8,9,10,11,12,37]. Therefore, we hypothesize that when *NIN* was recruited into root nodule formation, it came under the control of the common signaling pathway in order to express in nodules. Further, at a certain moment, it also lost its constitutive expression.

The spatiotemporal regulation of *NIN* has been studied in detail in Medicago and Lotus, and it is highly complex. We will first summarize this, and then discuss whether such regulation also occurs in other species. Upon inoculation, *NIN* is induced in the epidermis, where it is required for infection. This was demonstrated by *in situ* hybridization, promoter GUS reporter constructs and root hair transcriptome analyses [40,41,42,43]. The expression of *NIN* in the epidermis could be cell-autonomously regulated upon perception of NFs, which induce the common signaling pathway, resulting in the activation of the transcription factor CYCLOPS by phosphorylation [44]. It was first shown in Lotus that phosphorylated CYCLOPS binds to the *NIN* promoter in a sequence-specific manner to regulate *NIN* expression [44]. In Medicago, the CYCLOPS binding site is located about 3 kb upstream of the *NIN* start codon [41]. A 5-kb long promoter including the CYCLOPS binding site driving *NIN* expression can restore infection thread formation in a *nin* knockout mutant [41]. However, it cannot restore nodule organogenesis, showing that additional *cis*-regulatory elements are required for this (see below). Deletion of the CYCLOPS binding site dramatically reduced the infection thread-forming efficiency [41]; however, this site is not required for tight root hair curling, and bacterial colonies are still formed in these curls. This resembles the Lotus *cyclops* and Medicago *interacting protein of DMI3-2* (*ipd3-2*) mutant phenotype [45,46]. This indicates that, in addition to the CYCLOPS binding site, other *cis*-regulatory element(s) must be present, which is sufficient to induce *NIN* expression that leads to tight root hair curling. It was assumed that the CYCLOPS binding site results in a higher expression level of *NIN*, and the initiation of infection thread formation requires a higher threshold level than curling [41]. 

Interestingly, CYCLOPS binding site is conserved in legume *NIN* promoters [41], and it also occurs in the Parasponia *NIN* promoter (R. Huisman, personal communication [47]). However, they do not occur in the *MtNLP1* promoter. This suggests that the gaining of the CYCLOPS binding site occurred after the duplication that resulted in the NIN and NLP1 subgroups (Figure 1). This CYCLOPS binding site also does not occur in *NIN* orthologues of Arabidopsis (*NLP1/2/3*). Taken together, it can be hypothesized that the gain of the CYCLOPS binding site had occurred when *NIN* was recruited into the nodule formation process, and was present in the *NIN* of the ancestor of the NFC. This gain of the CYCLOPS binding site seems an essential step for its expression during nodule formation.

At an early stage of Medicago nodule development, when rhizobia have only colonized the epidermis, *NIN* expression and cell divisions are induced in the pericycle [41]. Subsequently, both extend to the inner cortex and endodermis [41]. Because NFs are immobile molecules [48], most likely a mobile signal was generated upon perception of NFs in the epidermis. This mobile signal was then translocated to the pericycle to activate *NIN* expression and cell division there [41].

The first insight that other, more remote, promoter elements are involved in the induction of *NIN* in inner layers came from studies with the *nin* weak alleles *daphne* (Lotus) and *daphne-like* (Medicago), in which the infection process is induced but nodule organogenesis is blocked [40,41]. *NIN* expression was shown to be induced in the epidermis in *daphne-like*, but not in pericycle [41]. Both mutants contain a mutation in the promoter region of *NIN* due to a chromosome translocation [40,41]. In both cases, these are located upstream of the CYCLOPS binding site [41]. This strongly suggested that the regulatory region required for *NIN* expression in the pericycle and *NIN*-controlled cell division is located upstream of the insertion. Recently, a remote *cis*-regulatory region named CE region was identified, which is essential for NIN-controlled nodule organogenesis and the expression of *NIN* in the pericycle [41]. This CE region contains many putative cytokinin response regulator (RR) binding sites [41]. Studies in which cytokinin is applied exogenously did not induce *NIN* expression in either *daphne* or *daphne-like* [40,41], supporting the conclusion that that this region is important for cytokinin-induced *NIN* expression. In addition, both the cytokinin receptor *CRE1* and B-type cytokinin response regulator *RR1* are expressed in the pericycle prior to *NIN* expression and cell division there [41]. Taken together, it is very likely that cytokinin signaling via the CE region induces *NIN* expression in the pericycle, and this leads to nodule organogenesis. It has been shown in Medicago that *YUCCA* genes, involved in auxin biosynthesis, are induced in an inner cell layer of the root (most likely pericycle) in a NIN-dependent manner [49]. Therefore, it is probable that subsequent cell divisions are induced by local auxin production.

The CE region is conserved in legumes and is located far upstream from the *NIN* start codon [41]. The distance varies between species. For example, in Medicago it is about 18 kb upstream, and in Lotus it is about 45 kb upstream of the *NIN* start codon [41]. Given that the CE region is conserved in legumes, it is very probable that the regulation of *NIN* expression in the inner layers by cytokinin signaling is conserved in legumes.

In Actinorhizal(-like) plants, NIN is required for nodule organogenesis, and this involves divisions in the inner root cell layers [12,50]. So is the CE region conserved in actinorhizal *NIN* genes? So far, only Parasponia *NIN* (*PanNIN*) has been analyzed, and the CE region was not identified (L. Rutten, personal communication [51]). Nodule formation in Parasponia is induced by Nod factors, and so, like in legumes, a mobile signal seems essential to trigger cell division in inner layers. Whether or not *NIN* is expressed in the Parasponia pericycle cells when cell division is induced has not been studied. However, this seems probable, as a putative target of PanNIN, namely *PanNF-YA1* (*NUCLEAR TRANSCRIPTION FACTOR Y SUBUNIT A-1*), is induced there [12]. *NF-YA1* has been shown to be a direct target of NIN in Lotus [52]. In Medicago, the expression of *NIN* and *NF*-*YA1* coincides in the dividing pericycle cells and other nodule primordium cells [41]. Further, *PanNF-YA1* expression has been shown to be PanNIN-dependent [12]. As the CE region does not occur in the Parasponia *NIN* promoter, the mobile signal most likely does not regulate *NIN* expression through cytokinin signaling. This conclusion is in line with a recent study in which cytokinin is applied to different plant species [53]. It was shown that exogenous application of cytokinin induces nodule-like structures on nodulating legume species, and in Lotus, for example, this depends on NIN [54]. However, nodule-like structures were not induced by cytokinin on either non-nodulating legumes or on actinorhizal species [53]. The non-nodulating legumes most likely have lost the ability to form cytokinin-induced nodule-like structures due to the loss of *NIN*, whereas in actinorhizal plants this might be due to the absence of a CE region in their *NIN* promoter. Therefore, the gain of cytokinin responsive elements in the *NIN* promoter seems specific to the legumes, and might not have already occurred in the ancestor of the NFC.

It cannot be excluded that more changes in the *NIN* promoter evolved during nodule evolution. For example, NSP1/2, IPN2 and ERN1 have been shown to play a role in Medicago and Lotus *NIN* expression [55,56,57,58,59]. However, it is not clear that this involved specific evolutionary adaptations in the promoter of *NIN*, related to its recruitment for nodulation.

Taken together, there seems to be (at least) three main evolutionary changes concerning the regulation of *NIN* expression during nodule evolution. First, it gained the CYCLOPS binding site, most likely in the ancestor of the NFC. Second, at an early stage of nodulation, the constitutive expression is lost by which, in nodulating plants, *NIN* became nodule-specific. Third, it developed cytokinin-regulated *NIN* expression during nodule organogenesis, which most likely occurred in the legume branches.

## 4. NIN-Controlled Epidermis–Pericycle Communication; A Conserved Module?

It seems probable that the activation of *NIN* in the inner root layers of legumes and Parasponia requires a mobile signal. What is the probability that this is a general property of nodulating plants? It has been proposed that the last common ancestor of the NFC formed a nodule symbiosis with *Frankia* [60]. Although it has not been demonstrated that *Frankia* can produce Nod factors, basal *Frankia* strains do have genes that are homologous to the rhizobial *nod* genes [61,62] that are involved in Nod factor biosynthesis. Therefore, it has been hypothesized that *Frankia*-induced nodulation in the last common ancestor of NFC by secreting Nod factors [60]. If so, as has been argued for legumes and Parasponia, a mobile signal might also be required to induce *NIN* in the inner root layers in the common ancestor, as well as in current actinorhizal plants that interact with *Frankia* that secrete Nod factors. It has been shown in legumes that the production of the mobile signal depends on *NIN* expression in the epidermis [41]. So *NIN*, expressed in the epidermis might also induce the formation of a mobile signal in the actinorhizal plants that interact with Nod factor-producing *Frankia*. Subsequently, that mobile signal triggers *NIN* expression and cell division in the pericycle. Thus, we hypothesize that, in legumes and these actinorhizal plants, a conserved module, involved in communication between the epidermis and pericycle, is active, and this was already present in the common ancestor of the NFC: Nod factor signaling induces *NIN* expression in the epidermis; NIN activates the production of a mobile signal; this mobile signal induces *NIN* expression and cell division in the pericycle. Is such a module maintained in all actinorhizal plants? Some Actinorhizal plants interact with *Frankia* strains that do not produce NFs, such as CcI3, which secretes a hydrophilic symbiotic signal that is resistant to chitinase [50]. Therefore, it is chemically distinct from the amphiphilic and chitin-based NFs. This symbiotic signal is able to induce *NIN* expression in the Casuarina root epidermis, and so NIN might still be able to induce the production of a mobile signal, which subsequently induces cell division in the pericycle. Alternatively, the mobile signal might not be required in these Actinorhizal plants if the symbiotic signal itself is mobile. However, it has been proposed that an immobile NF may facilitate the detection and localization of the bacteria by which the redirection of root hair growth is induced and a curl is formed that entraps the bacteria [48]. CcI3 infect Casuarina roots through infection threads [50]; therefore, it seems likely that this symbiotic signal is immobile, like NFs.

In case cytokinin is the mobile signal in legumes, then actinorhizal plants will most likely form a different mobile signal. Alternatively, the nature of the mobile signal is conserved, and it induces cytokinin signaling in legumes and a different pathway in actinorhizal(-like) plants. The induction of the formation of a mobile signal might be an ancestral/original function of NIN in the epidermis. As described above, NIN is essential for infection thread formation. Although it has not yet been studied, it seems likely that for more primitive infection modes like crack entry, NIN might not be required. Therefore, the ancestral function of NIN in the epidermis is probably the induction of the production of the mobile signal, whereas its role in infection thread formation evolved later.

## 5. Function of NIN: Acquired upon Recruitment or Adopted from NLP-Controlled Process?

NIN is essential for nodule initiation, including infection thread formation, nodule organogenesis and AON. We will discuss the functions of NIN in these different processes, including some of its target genes. Further, we will discuss to what extent they are adopted from processes in which NLPs function, or whether they are innovations after its recruitment.

### 5.1. NF-Ys

NF-Ys are transcription factors, and some of them play a role in the NIN-induced nodulation processes. NF-Y transcription factor complexes are composed of three subunits: NF-YA, NF-YB and NF-YC [63]. *NF-YA1* and *NF-YB1* have been shown to be direct targets of NIN in Lotus [52]. Knockout mutations in *NF-YA1* cause aberrant infection thread formation in Medicago [64] and a block of intracellular infection in Parasponia [12]. In *Phaseolus vulgaris* (common bean), the knockdown of *NF-YC1* also causes aberrant infection thread formation [65]. Further, NF-Ys play a role in nodule organogenesis. In Lotus, Medicago and Parasponia, *NF-YA1* mutations lead to the formation of small nodules and/or reduced nodule numbers [12,52,66]. Further, in both Medicago and Parasponia nodule primordia and nodules, *NF-YA1* is expressed at sites where *NIN* has been shown to be expressed [12,41]. This suggests that the involvement of a NIN-NF-Y module in intracellular infection, and nodule organogenesis is conserved within the NFC and it evolved at an early moment during nodule evolution. This raises the question of whether NF-Ys are involved in NLP-controlled processes (in species outside NFC), and whether these processes are related to nodule organogenesis and/or intracellular infection.

In Parasponia [12], a mutation in *NF-YA1* also has a non-symbiotic phenotype, namely, reduced lateral root formation. Moreover, in Lotus, overexpression of *NIN* or *NF-YA1* induces extra cell division in the pericycle, and this leads to the formation of lateral roots with a malformed tip [52]. In addition, overexpression of *NF-YA1* and *NF-YB1* in Lotus increases lateral root densities to twice those of empty vector controls [67]. This suggests that an ancestral function of NF-YA1 could be related to lateral root formation, a process very similar to nodule organogenesis. This hypothesis is further supported by studies on the *NF-YA1* orthologues in Arabidopsis, as AtNF-YA2/AtNF-YA10 are both involved in primary and lateral root growth [68]. In addition, they are expressed in pericycle cells, where lateral root as well as nodule organogenesis are initiated.

Interestingly, transcriptome analysis of the Arabidopsis *nlp7* (from a different orthogroup than *NIN*) mutant shows altered expression levels of several genes encoding NF-YA subunits, including *AtNF-YA2* and *AtNF-YA10* [69,70]. This suggests that (some) NLPs can regulate the expression of *NF-Y (A1)*. So, the NIN-NF-Y module might have been adopted from a NLP-NF-Y module which already occurred before nodulation evolved. 

How can NF-Y be involved in both infection and nodule organogenesis? In mammals, NF-Y regulates the expression of cell cycle genes [71]. During root nodule development, not only nodule organogenesis requires expression of the cell cycle genes, but the passage of infection threads through cells also requires entry into the cell cycle [42]. Therefore, NF-Y might regulate infection, as well as organogenesis, through the activation of cell cycle genes. This is supported by the knockdown of NF-YC in the common bean, which results in the reduced expression of cell cycle genes, whereas overexpression of *NF-YC1* causes higher expression levels of these genes [65]. Further, in Lotus, ectopic expression of *NIN*, or *NF-YA1* and *NF-YB1*, results in ectopic expression of a cyclin gene [52].

Accumulation of the phytohormone auxin is correlated with mitotic activity. For example, when lateral root formation is initiated, auxin accumulates in the pericycle cells [72]. In nodulation, auxin signaling is required for both nodule organogenesis and the initiation of infection threads [73,74]. Further, it has been shown that NF-YA1 directly regulates the expression of *STY* genes, which encode transcription factors that regulate the expression of *YUCCA* auxin biosynthesis genes in Arabidopsis [75,76,77,78]. So, NF-Y might, by regulating auxin biosynthesis genes, stimulate entry into the cell cycle, which is involved in both infection and organogenesis, and this also holds for NIN, which regulates *NF-Y* expression.

### 5.2. LBD16

Recent studies show that NIN directly regulates the expression of *LBD16* (*lob-domain protein 16*), which is known to be essential for lateral root formation [49,67]. In Medicago, an *lbd16* mutant shows similar defects in nodule and lateral root initiation [49]. Like NF-YA1, LBD16 promotes auxin biosynthesis via transcriptional induction of *STY* and *YUCCA* genes [49]. In Lotus, it was shown that LBD16 and NF-Y in an additive way regulate nodule organogenesis, as a mutation in *LBD16* enhances the nodulation phenotypes of *nf-y* subunit mutants [67]. In addition, the co-expression of *LBD16* and *NF-Y* subunit genes can partially replace *NIN*, as it can rescue nodule organogenesis in the weak *nin* allele *daphne* [67]. In Parasponia, the expression level of *LBD16* is about six times increased at the initial stages of nodule formation [6]. This level of induction is comparable to that in Lotus and Medicago [49,67]. In Arabidopsis, *LBD16* has been shown to be a direct target of NLP7 [70]. Therefore, NIN-controlled *LBD16* expression most likely is adopted from a non-symbiotic module in which its expression is controlled by NLP, and this module evolved before the occurrence of nodulation.

### 5.3. NPL1

Cell wall remodeling is required during rhizobial infection [15]. In Medicago root hairs, cell wall modification genes are induced upon rhizobial infection, and the expression of many of them depends on NIN [74]. Among these genes, only *NODULATION PECTATE LYASE 1* (*NPL1*) shows nodule-specific expression [74]. In both Lotus and Medicago, NPL1 is essential for infection thread formation [74,79]. Its expression is regulated by NIN, and it is highly induced at infection sites [74,79]. *NPL* seems to be the result of a tandem gene duplication, and its orthologues occur in *Glycine max* (Soybean), *Lupinus albus* (Lupin) and *Arachis ipaensis* [74]. So, the NIN-NPL module is most likely conserved in the papilionoid legume sub-family.

Parasponia has three putative *NPL* orthologs, namely *PanPLL8*, *PanPLL9* and *PanPLL10* [6]. However, the most closely related *PanPLL8* is not nodule-specifically expressed, and *PanPLL9* and *PanPLL10* are only four-fold induced at the initial stages of colonization. In later developmental stages, the relative fold change is even lower [6]. This is different from legumes, in which *NPL1* is highly induced [74,79]. This suggests that the NIN-controlled nodule-specific expression of *NPL* might have been gained within the legume branch.

### 5.4. RPG

*Rhizobium-directed polar growth* (*RPG*) is a gene of which the expression is controlled by NIN [74]. RPG is a long coiled–coil protein that is nuclear-localized [80]. In Medicago, it has been shown to be essential for normal root hair curling and infection thread formation [80]. RPG is nodule-specifically expressed in Medicago, Lotus and Parasponia [6,80,81], which is consistent with the loss of a functional *RPG* gene in several non-nodulating NFC species [5,6]. This supports that it only has a symbiotic function in the NFC. In Lotus, it has been shown by ChIP-seq that *RPG* is a direct target of NIN [13,74]. Together, this suggests that a NIN-controlled *RPG* expression evolved early in the NFC, or it was already controlled by an NLP before nodulation evolved. However, neither NLP7 CHIP-chip nor the *nlp7* transcriptome indicated that the expression of an *RPG* homologue is controlled [69,82]. To further support that NIN-controlled RPG expression evolved early in NFC, the RPG regulation in other *nlp* mutants, especially of *NIN* orthologues, remains to be analyzed.

### 5.5. CLEs

As described above, AON is essential for the regulation of nodule numbers, to balance the gains and costs of nodulation. In Medicago and Lotus, the *CLE* genes that are involved in AON are nodule-specifically expressed in a NIN-dependent manner [13,32,74]. In addition, PanCLE9, which is the putative Parasponia orthologue of Medicago CLE12/13, is also expressed at enhanced levels in nodules [6]. This suggest that NIN-controlled AON is conserved in NFC. However, the nodule-specific expression of CLE genes is not only controlled by NIN, but also by NLP whereas in response to nitrate [35]. In Lotus, it has been shown that both NRSYM1 (one of the NLPs belonging to the same orthogroup as AtNLP7) and NIN can bind to the Lotus CLE-RS2 promoter, and NRSYM1 even has a higher affinity [35]. In addition, Arabidopsis *nlp7* mutants have altered expressions of several *CLE* genes, including *AtCLE5/6/7*, which are putative orthologues of *LjCLE-RS1/2* [69]. Therefore, the function of NIN in controlling AON/CLE genes expression is most likely adopted from an ancestral NLP module.

## 6. Are NIN and NLPs Functionally Equivalent?

Several of the processes and genes regulated by NIN appear to be adopted from those controlled by NLPs. Therefore, it seems probable that NIN and NLPs are to some extent functionally equivalent. To test this, *MtNLP1*, the closest Medicago *NIN* paralogue, and *AtNLP1*, an Arabidopsis *NIN* orthologue driven by the Medicago *NIN* promoter, were introduced into the Medicago *nin-1* knockout mutant (J. Liu, unpublished data [83]). However, neither the nodules nor infection threads were formed. This shows that NIN and these two NLPs are not functionally equivalent. So their protein sequence has diverged, and most likely NIN obtained amino acid changes that are essential to its function in nodulation.

Most of the NLPs that have been studied are located in the cytoplasm under nitrate starvation, and when high nitrate is sensed, they are transported to the nucleus [35,36,38,82,84]. In contrast, in Medicago nodules, NIN is located in the nucleus of all the cells in which it is expressed, and this became independent of nitrate sensing (Liu et al., unpublished data [85]). So, a major difference between NLPs and NIN is the constitutive nuclear localization of NIN, and this might be the reason that NLPs cannot complement the *nin* mutant.

In Arabidopsis, it has been shown that, upon nitrate sensing, serine 205 (S205) in AtNLP7 becomes phosphorylated, after which it is translocated from the cytoplasm to nucleus [86]. In addition, S205 in AtNLP7 is located in the region previously predicted as NLP-specific, and it is absent in Medicago and Lotus NIN, whereas MtNLP1 and AtNLP1 have it [39,86,87].

This serine might be the reason that MtNLP1/AtNLP1 fails to complement *nin-1*, as nodulated plants are grown under low nitrate (0.5 mM) conditions. However, phosphomimic versions of MtNLP1/AtNLP1, serine-modified into aspartate, also did not complement *nin-1* (J. Liu, unpublished data [83]). This indicates that in addition to this serine, other changes are introduced in NIN that are required for it to be functional in nodulation. Identifying such critical changes will provide insight into the evolution of NIN.

## 7. Concluding Remarks

Both functional and phylogenomic analysis underpins the conserved key position of NIN in root nodule symbiosis. Further, the recruitment of NIN in the nodulation process seems to be the key step in the birth of this symbiosis. Therefore, understanding which minimal changes were required will provide major insights into the evolutionary trajectory leading to nodule formation. NIN is closely related to NLPs with whom it shares some downstream targets, pointing to partial functional equivalence. Two major changes are the regulation of expression by the microbe via the common signalling pathway, and the constitutive nuclear localization. Further adaptations will have occurred within the NFC. An example is the regulation of expression by cytokinin in the legume branch. Further studies on NIN orthologues inside and outside of NFC will help to refine the picture. The puzzle in understanding the evolution of NIN comes from its complicated spatiotemporal expression pattern and its multifunctionality. Until now, almost all studies on NIN have focused on its role during nodule initiation. However, *NIN* is also expressed in the mature nodule [10,88], where it most likely plays a different role than it does during the initial stages. Further studies on NIN in later nodule developmental stages will complete the picture of the evolution of NIN.

## Figures and Tables

**Figure 1 genes-11-00777-f001:**
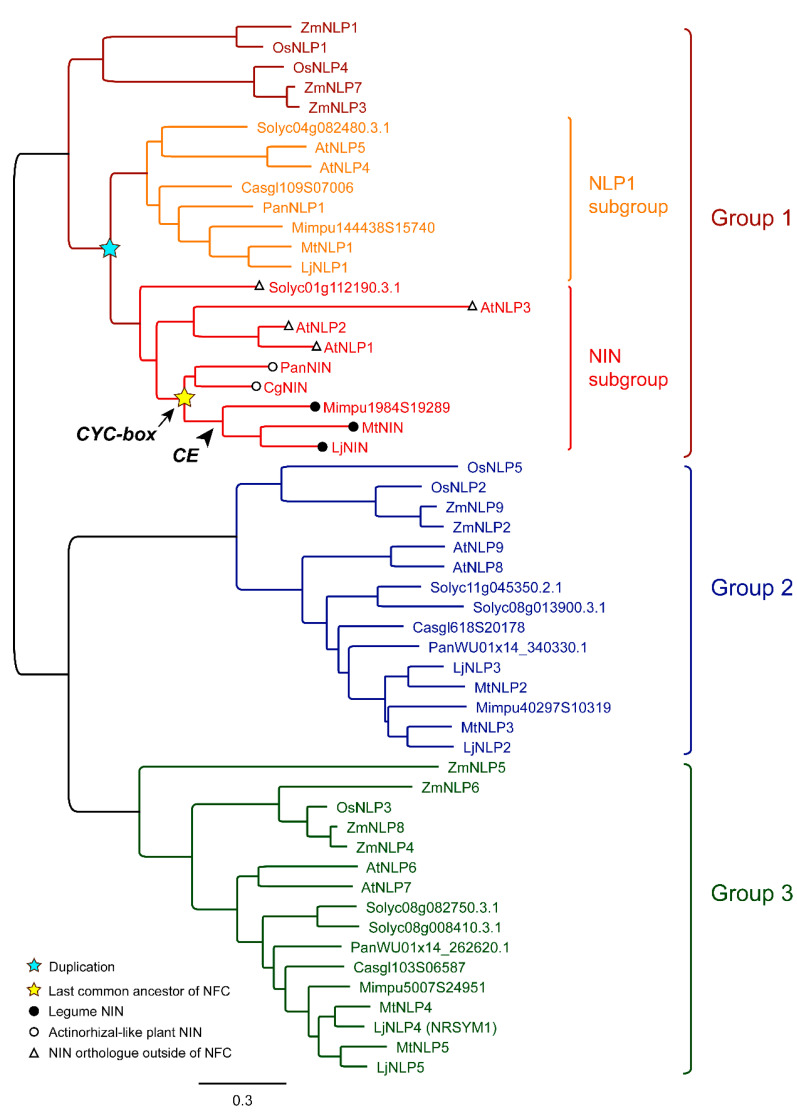
Phylogenetic tree of NIN and NLPs. The tree comprises 53 NIN/NLPs from *Zea mays* (Zm), *Oryza sativa* (Os), *Solanum lycopersicum* (Solyc), *Arabidopsis thaliana* (At), *Casuarina glauca* (Casgl/Cg), *Parasponia andersonii* (Pan), *Mimosa pudica* (Mimpu) *Medicago truncatula* (Mt) and *Lotus japonicus* (Lj). These NIN/NLPs are divided into three orthogroups as indicated. Dicots of Group 1 most likely undergo duplication (blue star), which generates NIN and NLP1 subgroups. The NIN subgroup comprises symbiotic NIN from NFC species (yellow star indicates the last common ancestor of NFC), including NIN of actinorhizal-like plant NIN (hollow circle) and legume NIN (filled circle), and putative NIN orthologues in non-nodulating species outside of NFC (hollow triangle). The CYCLOPs binding site (*CYC-box*) in the *NIN* promoter was likely gained by the ancestor of NFC (arrow), whereas cytokinin responsive elements (CE) in the *NIN* promoter specifically evolved in the legume branch (arrow head). Corresponding accession numbers and protein sequences are listed in the Appendix A.

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
