# Peer review of "Evolution of NIN and NIN-like Genes in Relation to Nodule Symbiosis"

_genes, 2020, doi:10.3390/genes11070777_

Round 1

Reviewer 1 Report

This manuscript is very interesting, well organized and written. This work should have a significant contribution to the field.

Author Response

Thanks for your kind words and comments.

Reviewer 2 Report

In the manuscript “Evolution of NIN and NIN-like genes in relation to nodule symbiosis”, the authors reviewed the role of NOUDLE INCEPTION (NIN) during nodulation in legumes, nonlegume Parasponia andersoniiand actinorhizal plants. They also discussed the evolution of NIN and NIN-like proteins (NLP) in relation to nodule symbiosis. The paper is well-organized and easy to follow, however I find the following issues that should be resolved before publishing this paper.

Major points:

  1. The nomenclature used in this review is incorrect and confusing, with interchange between the latin name versus the common name. Please be consistent in how species are referred throughout the review.  The binomial names of species should be usually typeset in italics.

Minor points:

  1. Page 2, line 47-49, please combine these two sentences into one sentence.
  2. Page 2, line 86, “costs and benefit” should be “costs and benefits”. The references cited here are all related to autoregulation of nodulation in Lotus japonicus. Please cite some work from other legume species as well.
  3. Page 2, line 88, “send” should be “sent”. Please provide the full name of “CLE” at first use.
  4. Page 2, line 89, “this” should be “these”.
  5. Page 2, line 91, “send” should be “sent”.
  6. Page 3, line 99, “NIN is partially functionally equivalent with NLPs” is also supported by Lin et al., Nature Plants, 2018 which demonstrates that NIN interacts with NLPs to mediate nitrate inhibition of nodulation in Medicago truncatula.
  7. Page 3, line 100-101, please provide the reference for the presence of NIN orthologues in tomato and Arabidopsis thaliana.
  8. Page 3, line 104, “Actinorhizal-like” should be “actinorhizal-like”.
  9. Page 3, line 108, “mono-” should be “monocot”.
  10. Page 3, line 115-117, the point that NINis lost in non-nodulating species is supported by the genome sequencing data. Therefore it should refer to the gene NINnot the protein NIN.
  11. Page 5, line 156, please rephrase “it still complements for tight root hair curling”.
  12. Page 5, line 157, “curl” should be “curls”. Please provide the full name of “ipd3” at first use and delete “(cyclops)”.
  13. Page 5, line 164, “the gain of the” should be “gain of the”.
  14. Page 5, line 179, the description of daphnemutant in Lotus japonicusis not correct. The nature of daphnemutation is a reciprocal chromosomal translocation between chromosome II and III, not an insertion of chromosome II sequences into chromosome III.
  15. Page 6, line 190-192, Schiessl et al., Current Biology, 2019 shown that the auxin biosynthesis regulator STYLISH(STY) is also activated in a NINand LBD16-dependent manner.
  16. Page 6, line 194-197, I am confused by the logics in this paragraph. Please explain why the presence of CE region far upstream from the NINstart codon results in the cytokinin-regulated NIN
  17. Page 6, line 213-215, I can’t agree with the statement here. Cytokinin-induced pseudonodule formation is a part of its physiological effects. Non-nodulating species do respond to cytokinin treatment in a dose-dependent reduction in either root length or lateral root number or both (Gauthier-Coles et al., Frontiers in Plant Science, 2019).
  18. Page 6, line 231-234, currently there is no evidence supporting that Frankiacan produce Nod factors. The Frankiaroot hair deforming factor has been demonstrated to be structurally different from rhizobial Nod factors or mycorrhizal Myc-LCOs (Cérémonie et al., Canadian Journal of Botany, 1999 and Cissoko et al., Frontiers in Plant Science, 2018). I think it doesn’t make sense to discuss a hypothesis which is not well supported.
  19. Page 7, line 259-260, please provide references for “NIN is not required for crack entry infection”.
  20. Page 7, line 269, please provide the full name of “NF-Y” at first use.
  21. Page 7, line 283, the description of nf-ya1mutant in Lotus japonicusis not consistent with the conclusion that “nf-y mutations did not influence lateral root densities” in the cited reference
  22. Page 8, line 324, please rephrase “this module evolved before nodulation evolved”.
  23. Page 8, line 328, the correct nomenclature of NPLgene is NODULATION PECTATE LYASE.
  24. Page 8, line 330, here NPLis mentioned as a gene, it should be in italics.
  25. Page 9, line 338, “NPL” should be “NPL”.
  26. Page 9, line 346-347, “CHIP” should be “ChIP”. The conclusion “In Lotus, it has been shown by CHIP-seq that RPG is a direct target of NIN” is not supported by Liu et al., Plant Physiology, 2019.
  27. Page 9, line 368-370, please provide references for this conclusion.
  28. Page 9, line 379, “Serine” should be “serine”.
  29. Page 9-10, line 384-385, please provide references for this conclusion.

Reviewer 3 Report

This review manuscript has been written very well but needs minor improvement.

L283; Please confirm if Lotus nf-y mutants display the lateral root phenotype or not.

L380-381; The authors descrived that S205 is conserved in NLP, but not in Lotus and Medicago NIN. How about NIN orthologues in Parasponia and actinorhizal plants? It may be an important point for considering the functional evolution of NIN proteins.

Please cite papers at several points.

L162-163 (for Parasponia)

L199-200

L322

L368-370

L385
